# Inhibition of Cancer Development by Natural Plant Polyphenols: Molecular Mechanisms

**DOI:** 10.3390/ijms241310663

**Published:** 2023-06-26

**Authors:** Alexander Lyubitelev, Vasily Studitsky

**Affiliations:** 1Biology Faculty, Lomonosov Moscow State University, 119234 Moscow, Russia; varanus.storri@gmail.com; 2Fox Chase Cancer Center, Philadelphia, PA 19111, USA

**Keywords:** polyphenols, cancer, chemoprevention, mutation, antioxidant, epigenetics, inflammation, microbiota

## Abstract

Malignant tumors remain one of the main sources of morbidity and mortality around the world. A chemotherapeutic approach to cancer treatment poses a multitude of challenges, primarily due to the low selectivity and genotoxicity of the majority of chemotherapeutic drugs currently used in the clinical practice, often leading to treatment-induced tumors formation. Highly selective antitumor drugs can largely resolve this issue, but their high selectivity leads to significant drawbacks due to the intrinsic tumor heterogeneity. In contrast, plant polyphenols can simultaneously affect many processes that are involved in the acquiring and maintaining of hallmark properties of malignant cells, and their toxic dose is typically much higher than the therapeutic one. In the present work we describe the mechanisms of the action of polyphenols on cancer cells, including their effects on genetic and epigenetic instability, tumor-promoting inflammation, and altered microbiota.

## 1. Introduction

Malignant tumors are among the longest-known diseases in the mankind history [1]. The most significant progress in cancer research was made during the last century and a half, and today, most of the cancers are considered as a set of acquired genetic disorders that lead to uncontrolled proliferation of the affected cells [2,3]. Despite the significant progress that was achieved in this field, malignant tumors continue to be one of the most important sources of morbidity and mortality throughout the world [4]. One of the most important causes of this situation is significant inter- and intra-tumor genetic heterogeneity. This heterogeneity arises from genetic differences between individuals and from different trajectories of acquiring malignant properties. This variety was described as the hallmarks of cancer; the list of these hallmark properties initially consisted of 6 positions^5^. Later decades confirmed the applicability of this concept [5,6], and the number of the hallmark properties and cancer-enabling characteristics grew up to 12^5^. Cancer-enabling characteristics, such as genetic and epigenetic instability, chronic inflammation, and altered microbiota, do not comprise malignant phenotype per se, but they may accompany the entire process of tumorigenesis, facilitating malignant transformation.

Acquiring all of these properties requires many large-scale changes in cellular physiology and genetics, which, with the exception of rare cases of chromotrypsis [7], cannot be made as a result of a single mutational event. Instead, malignant transformation happens as a result of a gradual accumulation of mutations during several years, or even decades [8]. This process includes three stages, called initiation, promotion, and progression [9]; each stage corresponds to a different type of lesion formed by affected cells: microscopic tumor at the stage of initiation, benign tumor with well-defined boundaries at the stage of promotion, and an invasive malignant tumor at the stage of progression [10]. At the stage of initiation, the first genetic alteration is acquired by the affected cells, providing them with an initial proliferative advantage. The stage of promotion is characterized by preneoplastic cell proliferation and by accumulation of additional genetic alterations. The stage of progression concludes the formation of a malignant tumor; it is characterized by constant and aggressive cell proliferation, invasiveness, and the ability to metastasize. The exact trajectory of acquisition of all the hallmark properties may not be identical for all cells within the neoplastic lesion, resulting in an intra-tumor genetic heterogeneity. Formation of metastases also contributes to the genetic heterogeneity of malignant tumors in a single organism, as metastases usually form at the distant sites from the primary tumors, where they are exposed to new factors that could be absent in their initial microenvironment [11]. This sequence of events only approximately describes the course of malignant transformation. Thus, in some cases, the metastases start to form long before the malignization process is concluded [12]. A single mutation is not always responsible for the emergence of a single hallmark property: it was established that as few as three mutations were sufficient for malignant transformation, at least for certain tumors [9,13]. Currently, it is considered that the number of tumor-promoting mutations in tumor cells at the early stages of carcinogenesis is relatively small; but as the disease progresses, the number grows accordingly. As the progressive deregulation encompasses more genes and processes in the cell, affected signaling pathways that are involved in supporting the cancer hallmark properties gradually become functionally redundant.

Processes responsible for the development of a malignant tumor are controlled by a small set of mutations for only a limited period of time. During the rest of the malignant transformation, its progression is driven by a broad spectrum of factors that simultaneously affect a multitude of cellular components and processes. At the same time, a number of anticancer drugs that are currently used and developed are characterized by a high specificity to their targets, and, although these drugs are able to slow tumor growth or even cause a significant reduction in its volume, their efficiency decreases drastically during long-term use. Such a decrease is caused by intra-tumor genetic heterogeneity, as the small number of malignant cells that are insensitive to the selected drug already exist within the tumor at the beginning of the therapy. Even a combination of several highly specific anticancer drugs cannot always prevent the emergence of this effect. Development of anticancer drugs with multiple molecular targets was proposed in order to overcome this drawback [14,15]. Another group of anticancer drugs that are currently in use in clinical practice consists of substances capable of damaging cellular DNA or inhibiting cell division. While their activity is not limited to a certain molecular target, they lack the selectivity for cancer cells, and the alterations of the genome of normal cells caused by these substances often lead to the formation of therapy-induced tumors.

One approach to reduce the cancer-caused mortality is to prevent the development of a malignancy rather than trying to cure it. It is established that around 45% of cancer cases are preventable [16]. The most straightforward approach to cancer prevention is avoidance of exposure to the environmental carcinogenic factors. This approach, in different forms, is applied in many countries around the world. However, not every environmental factor that is potentially carcinogenic, can be effectively avoided, either because of its ubiquitous nature, or because this particular factor is not yet identified as carcinogenic. Moreover, only a limited fraction of mutations that drive the process of carcinogenesis can be attributed to the environmental factors, with an average of around 29% of driver mutations that are caused by the external sources [17].

Cell-intrinsic processes might also be the cause of cellular DNA damage. Aside from mistakes made by replicative DNA polymerase during each S-period, there are also a multitude of sources of highly reactive compounds within the cells, such as reactive oxygen species [18]. To effectively mitigate DNA-damaging activity of both intrinsic and environmental factors, preventive measures should last for a long period of time, as malignant transformation and formation of metastases take years or even decades to conclude. The concept of cancer chemoprevention implies that such mitigation can be achieved either by neutralizing the DNA-damaging factors with chemopreventive drugs, thus abrogating initiation of tumorigenesis, or by preventing accumulation of the changes in cellular physiology, resulting in deceleration or even halting the promotion and progression stages of malignant transformation [19]. Chemopreventive compounds that abrogate the initiation step of malignant transformation are referred to as blockers. Agents that impede the transition of a pre-neoplastic lesion through the promotion and progression stages are known as suppressors. Chemopreventive compounds differ not only by their mechanisms of action, but also by the stage of the malignant transformation at which they act. Based on this criterion, these agents are classified into primary, secondary, and tertiary [20]. Primary chemopreventors suppress the formation of tumors, secondary chemopreventors are aimed at suppressing the transition of a benign pre-neoplastic lesion into malignant one, while tertiary chemopreventors reduce the risk of tumor recurrence after the successful therapeutic intervention. Thus, a potential chemopreventive drug should be suited for a long-term application, be well tolerated by the human organism, and should have a high toxic-to-therapeutic dose ratio.

When discussing anticancer chemotherapeutic and chemopreventive drugs, their cost is also a concern. Indeed, the annual cost of a chemotherapeutic anticancer drug may be as high as USD 100,000 annually [21]. For chemopreventive drugs, which are typically used for a long period of time, the problem of cost efficiency is of even higher relevance.

In summary, duration and mechanism of malignant transformation dictate a long list of requirements for a potential anticancer or chemopreventive drug: it should be highly specific towards tumor cells while having low or no toxicity, a broad spectrum of intra-cellular targets in a tumor, and high cost efficiency. At the first glance, these requirements cannot be achieved simultaneously. However, during the past decade, a dependence of cancer morbidity on diet was discovered: for individuals whose diet included a large amount of plant-based food, the incidence of cancer was lowered. Subsequent studies established that the main factor responsible for this correlation was the enrichment of dietary substances with plant polyphenols [22]. These compounds belong to the vast group of chemicals known as plant secondary metabolites, and are characterized by the presence of one or more aromatic rings and at least one hydroxyl group. Following the initial discovery of anticancer activity of plant polyphenols, it was found out that these substances can exert this activity by affecting multiple signaling pathways simultaneously. Their toxic concentration was also much higher than the therapeutic one. This combination of properties drew significant attention towards the possible use of plant phenolic compounds as anticancer and chemopreventive drugs. Here, we discuss the effects of polyphenols on cancer-enabling factors.

## 2. Structure and Properties of Plant Polyphenols

One of the most prominent features of plant biochemistry, which is almost completely absent in the animal kingdom, is the ability of plants to synthesize a broad spectrum of small organic molecules of multiple different structures, which are collectively known as plant secondary metabolites [23]. More than 200,000 different members of this group are currently known [24]. Secondary metabolites carry out a variety of functions, which include protection from ultraviolet radiation, regulation of physiological processes, resistance to physical stress, and interaction with different organisms, such as herbivorous animals or parasitic fungi [25]. Plant polyphenols are one of the largest groups of secondary metabolites—it includes around 8000 compounds with different structures [26]. Polyphenols may vary significantly by their chemical structure and molecular weight, but they always include at least one aromatic ring and hydroxyl group [27]. Several different variants of classification of the phenolic compounds were proposed. The most widely accepted of these variants divides plant phenolic compounds into two groups: flavonoids and non-flavonoids [28] (Figure 1). Flavonoids are derived from aromatic amino acids, and share a common tricyclic C6-C3-C6 structure [29]. Flavonoids are further divided into flavonols, flavones, isoflavones, flavanones, flavanols, anthocyanins, and anthocyanidins [30]. The group of non-flavonoid polyphenols includes phenolic acids, xanthones, stilbenes, tannins, and lignans [31] (Figure 1).

Polyphenolic compounds can be found in large quantities in fruits, seeds, and other edible parts of plants [32,33]. Fruits, berries, legumes, and other plant-derived foods and beverages [34], such as tea [35], wine [36], olive oil [37], or seasonings [38], are rich with polyphenols. Medicinal plants are another source of polyphenolic compounds [39]. The amount of polyphenols in a diet may differ drastically for different regions of the planet [40,41]. Epidemiological studies of the influence of a polyphenol-rich diet on human health suggest that the regular consumption of these compounds is associated with a lower incidence of a multitude of serious chronic diseases, such as metabolic syndrome [42], type II diabetes mellitus [43], neurodegenerative amyloidosis [44], cardiovascular diseases [45], and cancer [46].

As malignant transformation includes several stages and requires a long period of time to be completed [8], decrease in the incidence of cancer observed for individuals on a polyphenol-rich diet may be caused by the anticancer or chemopreventive properties of these compounds. Further experiments with tumor cell cultures and models, as well as clinical studies, established that polyphenolic compounds possess both of these activities [47,48]. For example, the treatment of different tumor cell cultures with plant phenolic compounds revealed the antiproliferative, antiangiogenic, cell cycle-arresting, and proapoptotic activity of these substances [49]. In particular, proapoptotic and antiproliferative activity of polyphenols was demonstrated for the MCF-7 breast cancer cell line [50,51], as well as for several prostate cancer cell lines [52]. Antiangiogenic activity of plant phenolics was demonstrated for murine [53] and human [54] healthy and tumor cell lines treated with these compounds. Anticancer and chemopreventive activity of polyphenols was confirmed by the results of in vivo experiments involving animal-induced tumorigenesis and tumor xenografts [55,56]. Treatment with polyphenols lowered the incidence of neoplastic lesions after initiation of tumorigenesis by N-nitrosobis(2-oxopropyl)amine [57] and suppressed the development of tumors stimulated by 1,2-dimethylhydrazine [58] and azoxymethane [59]. A similar treatment of the animals carrying tumor xenografts led to a significant reduction in tumor volumes and tumor growth rates [60,61]. Finally, the effectiveness of plant phenolic compounds as potential anticancer and chemopreventive drugs was confirmed by several clinical studies [62,63]. Phenolic compounds have antitumor activities not only alone [64,65], but also in combinations with other anticancer drugs [66,67]. At the same time, there are clinical studies in which plant polyphenols failed to demonstrate any anticancer or chemopreventive activity [68,69]. Some studies suggest that polyphenols might even facilitate tumorigenesis [70].

Confirmation of the anticancer and chemopreventive properties of polyphenols brought forth a question about the exact molecular mechanisms that underlie these activities. Unlike many other clinically used plant secondary metabolites that can interact with a limited number of molecular targets, or even with only a single one, plant phenolic compounds are able to interact with a broad spectrum of biological macromolecules of several different classes [71]. It was demonstrated that the anticancer activity of polyphenols might be determined by their specific interaction with signaling [72], catalytic [73], regulatory [74], and receptor [75,76] proteins, as well as by their non-specific chaperone activity [77]. The ability of phenolic compounds to specifically accumulate in cellular membrane lipid rafts, thus causing changes in cell surface receptor protein expression, were also demonstrated [78]. Nucleic acids are yet another molecular target of polyphenols. Polyphenols are not only able to bind to dsDNA, acting as intercalators [79], or major and minor groove ligands [80], but they also may interact with nucleic acid having secondary structures, such as tRNAs [81] or guanine quadruplexes [82].

Evidence of the biological activity of plant phenolic compounds accumulated to date, combined with data regarding their interaction with cellular targets, makes it possible to consider these substances as a basis for the development of perspective anticancer and chemopreventive drugs. Nevertheless, mostly non-specific binding of polyphenols to their molecular targets and a broad range of these targets make it necessary to extensively study the influence of polyphenols on regulatory pathways and processes involved in tumorigenesis and the development of cancer-enabling characteristics.

## 3. Genetic Instability

Human cells are exposed to a wide variety of DNA-damaging substances, both exogenous and endogenous, which include polycyclic aromatic hydrocarbons [83], polycyclic aromatic amines [84], and a number of other genotoxic agents of various chemical natures [85,86]. Some of them are carcinogens per se, while others are procarcinogens, and may exert carcinogenic properties only after they underwent metabolic activation. Metabolism of procarcinogens, as well as the other xenobiotics, consists of three sequential stages: metabolic activation, conjugation, and excretion [87]. Aromatic hydrocarbon receptor AhR plays a key role in the activation of the xenobiotic metabolism by regulating expression of metabolic activation and conjugation enzymes [88,89] (Figure 2). Binding AhR to its ligand causes a conformational change, thus leading to exposure of a nuclear translocation signal, which is covered by molecules of chaperones, such as p23 or Hsp90, while AhR is in an inactive state. The nuclear translocation signal mediates transfer of the AhR-chaperone complex to the nucleus, where chaperones dissociate from AhR, and the receptor, in turn, forms a heterodimer with Arnt. In this heterodimeric state, AhR is able to bind xenobiotic response elements (XREs) on cell DNA, thus activating the expression of enzymes involved in xenobiotic metabolism. This receptor is also involved in regulating inflammatory response, reaction to hypoxia, and cell cycle progression [90]. Organic compounds with several aromatic rings, including plant polyphenols, are the main ligands of AhR. Direct interaction between AhR and polyphenols was demonstrated for many compounds of this group [91]. The binding of polyphenols to AhR can modulate activity of this receptor by several different mechanisms (Figure 2); the most extensively studied among these is direct competitive inhibition. It was demonstrated that a number of polyphenolic compounds, namely curcumin [92], quercitin [93], and resveratrol [94], together with flavone, apigenin, luteolin, and galangin [95], may prevent AhR activation by polycyclic organic ligands in cellular extracts. Effective inhibitory concentrations of polyphenols in these experiments ranged from 0.019 to 50 µM. Kaempferol, luteolin, apigenin, and fisetin were able to prevent dioxin-mediated activation of AhR-regulated CYP1A1 gene in Caco2 human colon carcinoma cell culture. At the same time, treatment of the same cell culture with quercitin, robinetin, morin, and taxifolin alone, or in combination with dioxin, did not cause any reduction in AhR-regulated genes expression. Moreover, expression of these genes was significantly increased, suggesting that these phenolic compounds act as AhR agonists [96]. It was also demonstrated that quercitin, naringin, hesperidin, and hesperitin were able to enhance luciferase gene expression, which was put under AhR transcriptional control in genetically engineered reporter H1L6.1c2 cell culture [97]. Authors noted that only polyphenols with five hydroxyl residues in their structure were able to act as AhR agonists, whereas phenolic compounds with a different number of hydroxyls acted as antagonists or showed no activity towards AhR. Analysis of the interactions between plant phenolics and AhR made possible the development of molecular models that can predict in vitro agonistic activity of certain types of polyphenols towards this receptor protein [98]. Nevertheless, the principles that govern type and intensity of AhR modulation by plant phenolic molecules are yet to be fully understood [99]. Our understanding of these principles is limited by the prominent structural diversity of the polyphenolic substances and by controversial experimental results: depending on experimental conditions, the same phenolic compound demonstrated opposite effects towards AhR.

Mechanisms of modulation of AhR activity by polyphenols are not limited to direct competitive inhibition or agonism (Figure 2). Phenolic compounds can also affect a number of stages of the AhR signal transduction pathway. It was demonstrated that galangin, luteolin, apigenin, flavone, kaempferol [100], curcumin [101], and resveratrol [94] in concentrations around 5–50 µM were able to prevent the nuclear translocation of AhR in MCF-7 and Hepa-1c1c7 cell cultures. It was also shown that treatment of Hepa-1c1c7 cells with polyphenols can prevent formation of an active AhR-Arnt heterodimer: this ability was also demonstrated for galangin [102], naringenin, kaempferol, apigenin [100], and epigallocatechin gallate (EGCG) [100,103]. These compounds block AhR-Arnt heterodimer by different mechanisms: EGCG and galangin prevent association of AhR and Arnt by blocking the dissociation of chaperones from AhR. Apigenin and kaempferol stabilize the ternary complex between AhR, Arnt, and chaperones, while naringenin, kaempferol, apigenin, and EGCG were able to prevent Arnt phosphorylation, which is necessary for dimerization. The binding of polyphenols to active AhR-Arnt dimer may also inhibit activation of AhR-regulated genes by preventing binding of the dimer to the XRE DNA elements: the ability of preventing binding of the dimer to the XREs in tumor cell cultures was demonstrated for resveratrol [104], curcumin [101], quercitin, kaempferol [93], and galangin [102]. Plant phenolic compounds may also prevent activation of AhR-responsible genes by inhibiting recruitment of coregulators, such as the ERα estrogen receptor. This ability was demonstrated for resveratrol [94], genistein [105], and kaempferol [106], but the extent of the effect of this inhibition on the efficiency of the activation of transcription is yet to be determined. It is also noteworthy that in some cases, treatment with polyphenols instead of inhibition caused an enhancement of certain stages of the AhR signal transduction pathway: thus, treatment of Hepa-1c1c7 cells with curcumin facilitated nuclear translocation of ligand-bound AhR [101]. At the same time, curcumin efficiently inhibits subsequent stages of AhR activation, such as AhR-Arnt heterodimer formation. The mechanisms of plant polyphenol action on the AhR signal transduction pathway are summarized in Table 1.

The most important proteins, the expression of which is regulated by the AhR signaling pathway, are several proteins of the cytochrome P450 family. The human genome contains more than 50 genes of these redox enzymes. These enzymes participate not only in xenobiotic activation, but also in fatty acid and cholesterol biosynthesis, the metabolism of steroid hormones and vitamins, and unsaturated fatty acid oxidation [107]. Reactions of the metabolic activation of small organic molecules, which are responsible for procarcinogen activation, are catalyzed by cytochromes of subfamilies 1, 2, and 3; CYP1A1, A2 and A6, CYP2A13, CYP2B6, CYP2C9, and CYP3A4 are the most important among them [108]. Plant polyphenols can modulate activity of proteins of this family not only by affecting regulation of their expression, but also by interacting directly with the molecules. Using a combination of spectroscopic methods and molecular modeling approaches it was established that a large number of natural flavonoids is capable of binding to the active site of cytochromesCYP1A1, 1A2, 1B1, 2C9 and 3A4 [109]. For the latter two isoforms it was also found out that inhibition of their activity by flavonoid molecules is carried out by both competitive and non-competitive mechanisms [110]. It is noteworthy that for some flavonoids, such as amentoflavone, apigenin, and galangin, their half-maximal inhibitory concentration measured during this experiment was significantly lower than their previously determined concentration in human plasma, which suggests that the inhibitory activity of these polyphenols is physiologically relevant.

Plant polyphenols are not only able to prevent DNA damage, but also facilitate their detection and repair. For example, treatment with natural phenolic compounds might affect expression of the γH2AX histone variant that marks double-stranded DNA breaks in interphase chromatin, thus leading either to repair of these breaks [111] or to sensitization of tumors to DNA-damaging therapeutic agents [112]. Treatment with these compounds might stimulate DNA repair activity in normal tissues and promote apoptosis in tumor cells [113]. The latter type of activity suggests that natural phenolic compounds may enhance the efficiency of DNA-damaging anticancer therapy. Recent experimental results confirm the higher efficiency of this combined therapy [114].

Organic and inorganic free radicals are other important classes of DNA-damaging substances [115,116]. These compounds are not only capable of damaging DNA molecules by themselves, but they may also react with procarcinogens, thus activating them [117]. The main sources of free radicals in the cell are metabolic reactions and exposure to electromagnetic radiation. They may also form as by-products of the reactions catalyzed by ions of transition metals, such as iron and copper [118]. Free radicals can also damage cellular proteins, yielding them unable to carry out their functions [119]. As formation of the free radicals is unavoidable, living organisms had to develop effective measures to counteract their deleterious effects, such as redox enzymes and their cofactors, which can react with free radicals to neutralize them [120]. Numerous hydroxyl groups and certain other structural features determine the ability of plant polyphenols to react with free radicals, thus effectively scavenging them [121,122]. There are four known types of these reactions: three of them are based on proton and electron transfer from phenolic molecules to free radicals by different mechanisms; the fourth type includes adducts formed after interaction between polyphenol and a free radical [123]. Catechins and their derivatives, as well as plant extracts containing these substances, have the ability to effectively neutralize organic radicals [124] and reactive oxygen and nitrogen species [124,125] in vitro. EGCG has antioxidant activity in vivo, significantly reducing the effect of the induction of inorganic radical formation by sodium fluoride [126]. Gallic acid was able to reduce the severity of dimethylhydrazine-induced oxidative stress in rats, thus preventing induction of colon carcinoma by this carcinogenic compound [127]. Quercitin has a similar chemopreventive activity; it works by effectively mitigating the oxidative stress after induction of prostate carcinoma by high doses of testosterone in Sprague–Dawley rats [128]. Phenolic compounds from an extract of green tea leaves were able to increase the antioxidant potential of human plasma [129,130]. According to numerous clinical studies, consumption of anthocyanins and anthocyanin-rich dietary substances led to a significant decrease in the concentration of oxidative stress biomarkers, such as malonyldialdehyde and oxidized low-density lipoproteins [131]. Increase in the activity of gluthatione peroxidase and superoxide dismutase, as well as a general increase in antioxidant capacity, was also observed. These effects were more prominent for patients with diseases than for healthy volunteers.

Plant polyphenols can mitigate the oxidative stress not only by free radical scavenging, but also by preventing their formation. It was established that phenolic compounds can act as chelators of metal ions, thus decreasing the rate of free radical formation catalyzed by these ions. This kind of activity was demonstrated for flavonoids [132] and for non-flavonoid plant phenolic compounds, including phenolic acids, the structures of which include only a single aromatic ring [133].

Despite the significant amount of evidence of the antioxidant activity of plant polyphenols obtained during the experiments with cell-free extracts, cell cultures, and in vivo studies, the influence of natural phenolic compounds on the cellular redox status turned out to be more complex, and even prooxidant [134]. Polyphenols exerted their prooxidant activity in high concentrations, under high pH values, or in the presence of ions of certain transition metals [135]. Paradoxically, this pro-oxidant activity may also result in an increase in cellular antioxidant defenses: under physiologically relevant concentrations of polyphenols, the amount of reactive oxygen species that are formed by these reactions is not sufficient to induce any significant DNA damage, but enough to elicit cellular mechanisms of protection against the oxidative stress [136]. It is noteworthy that EGCG, previously viewed as a potent antioxidant, demonstrated pro-oxidant activity in this research. It was also established that certain cultivated cancer cells with metabolism and signaling networks distorted by malignant transformation react to the treatment with polyphenols in the opposite way as normal cells. Instead of lowering the severity of oxidative stress and decreasing the concentration of free radicals, this treatment led to a decrease in proliferative activity and to an induction of apoptosis, as the concentration of free radicals was increased to cytotoxic values [135].

Antioxidant activity of plant phenolic compounds is not limited to direct radical scavenging. Polyphenols can also affect the redox status of the cell by attenuating metabolic processes involved in the generation and neutralization of free radicals. Nrf2 is one of the most important transcription factors responsible for activation of the expression of oxidative stress response proteins. Oxidative stress induces dissociation of the Keap1-Nrf2 complex, preventing its Keap1 ubiquitination-induced proteasomal degradation, and thus leading to an accumulation of active Nrf2 and to the activation of the transcription of Nrf2-regulated genes [137]. It was established that quercitin may prevent ubiquitination of Keap1, facilitating accumulation and activation of Nrf2 [138]. EGCG was able to enhance Nrf2 activity in lung tissue in rats, abrogating fluoride-induced oxidative stress [126]. Results of molecular docking suggest that this effect could be determined by the interaction between EGCG and Keap1, preventing this protein from binding to Nrf2 and mediating its proteasome degradation. Similar results were obtained during treatment of L02 human hepatocyte culture with gallic acid, where this phenolic compound mitigated tert-butyl hydroperoxide-induced oxidative stress by preventing formation of the Nrf2-Keap1 complex [139]. Apigenin also has the ability to facilitate accumulation of Nrf2 in the cell, and to prevent high-fructose diet-induced oxidative stress [140]. Molecular modeling of interactions between apigenin and Keap1 suggest that these molecules may form a stable complex that is unable to bind to Nrf2.

Plant polyphenols can also modulate the activity of other proteins involved in the maintenance of the cellular redox status. Green tea polyphenols have an inhibitory activity towards expression of NADPH oxidase subunits, thus attenuating production of reactive oxygen species. The ability of EGCG and theaflavin-3,3′-digallate to inhibit lipopolysaccharide (LPS)-induced induction of inducible nitric oxide synthase (iNOS) expression was also established [141,142].

When considering the antioxidant activity of polyphenols for anticancer and chemopreventive applications, it should be kept in mind that reactive oxygen species are involved in regulation of a number of cellular processes, and in turn, are subjected to strict regulation [143]. There is also evidence that the mode of action of natural antioxidants in vivo differs significantly from the mechanisms identified by in vitro experiments [144]. These concerns raised doubts about the efficiency of high doses of natural antioxidants as anticancer measures [145], especially in combination with the other therapeutic approaches, whose mechanisms of action are based on induction of free radical formation in cancer cells [146,147]. All these complications demand separation of the antitumor and chemopreventive applications of polyphenolic antioxidants [148].

## 4. Epigenetic Reprogramming

DNA damage imposed by different factors affects the expression and functions of cellular proteins in a straightforward manner—by changing DNA sequence and thus affecting activity of the encoded proteins or misregulating their expression. However, the activity of the regulatory elements of the cellular genome is regulated not only by DNA sequence, but also by a number of non-genomic hereditable markers, commonly known as epigenetic factors. These factors include methylation of DNA, methylation, acetylation, and other post-translational modifications of histones, histone variants incorporated into chromatin, and non-coding RNAs [149]. It was established that disorders of epigenetic regulation may lead to development of one or more cancer hallmark properties, marking epigenetic instability as a cancer-enabling characteristic [150,151]. For example, global DNA hypomethylation found in many cancer cells enhances transcriptional activity in a non-selective manner, whereas hypermethylation of regulatory sequences of tumor suppressor genes may lead to activation of a number of genes and thus facilitate the malignant transformation [152]. Disorders of epigenetic labeling of histone writers, readers, or erasers lead to changes in interphase chromatin compaction and activity and affect gene expression [153]. Chromatin structure and activity is also regulated by the inclusion of different nucleosome core [154,155] and linker [156,157] histone variants. Aberrant activity of microRNA may facilitate expression of pro-oncogenic signaling pathway proteins or inhibit expression of tumor suppressor proteins at a translational level [158]. Finally, guanine quadruplexes might represent yet another system of epigenetic regulation, as suggested by the reverse correlation between the number of folded quadruplexes and the degree of cellular differentiation [159].

Plant phenolic compounds affect a multitude of cellular processes involved in epigenetic regulation. EGCG might attenuate activity of DNMT DNA methyltransferases by downregulating their expression [160], and by inhibiting their catalytic activity [161] after binding directly to these proteins [162,163]. Attenuation of the activity of methyltransferases was detected in cultures of esophageal carcinoma (KYSE 150), colon carcinoma (HT29), prostate adenocarcinoma (PC3), and breast cancer (MCF-7 andMDA-MB-231) cell lines, as well as for tumor xenografts, and led to reactivation of tumor suppressor genes silenced by hypermethylation. Curcumin was also able to rescue the expression of tumor suppressor genes by reversing their hypermethylation. Treatment of cultures of lung (A549 and H460) [164] and breast (MCF-7) [165] cancer cells with this phenolic compound led to a decrease in DNMT3b and DNMT1 methyltransferase expression, respectively. It is noteworthy that the inhibition of hypermethylation was detected at lower concentrations of this polyphenol, starting from 10 µM [166]. Treatment with quercitin potentiated the inhibitory activity of curcumin towards DNA methyltransferases of PC3 and DU145 prostate cancer cells: simultaneous treatment with both compounds inhibited the methyltransferase activity more efficiently than treatment with either substance separately [167]. Other plant polyphenols may also affect the DNA methylation. It was found that resveratrol, applied to MDA-MB-157 breast cancer cell culture in combination with pterostilbene, can decrease the levels of methylation of certain regions of genome by inhibiting the recruiting of DNA methyltransferases by STRT1 histone deacetylase [168]. Resveratrol was also able to selectively reverse hypermethylation of regulatory regions of tumor suppressor genes in MDA-MB-231 breast cancer cell culture [169]. Similar activity towards the RASSF-1α tumor suppressor gene was registered during clinical studies [170]. Remarkably, without pterostilbene, resveratrol has antiproliferative activity towards the MCF10CA1a breast cancer cell culture by the opposite mechanism: treatment with this phenolic compound led to recruitment of DNMT3b methyltransferase for the regulatory sequences of oncogenes, thus attenuating their expression by hypermethylation. Genistein also exerted suppression of DNA methyltransferase activity towards renal (A498, ACHN) [171], prostate (LNCaP, PC3) [172], and breast (MDA-MB-468, MCF-7) cancer cell lines; for the latter cultures, the inhibitory concentration of this phenolic compound was as low as 3.125 µM [173].

Plant polyphenols can also modulate activity of proteins involved in the epigenetic modification of histones in tumor cells. It was established that treatment of LNCaP [174] prostate and RKO, HCT-116 иHT-29 [175] colon cancer cell cultures with EGCG inhibited expression of HDAC 1-3 histone deacetylases. Treatment of HeLa cell culture with this polyphenol did not inhibit histone deacetylases, but instead resulted in the inhibition of histone acetyltransferase activity [176]. Curcumin-inhibited histone deacetylases, as treatment of Raji lymphoblast and several medulloblastoma cell cultures with this polyphenol, suppressed their proliferation by downregulating expression of histone deacetylases [177,178]. Quercitin also exerted modulatory activity towards both histone acetyltransferases and deacetylases, as treatment of HL-60 human leukemia cell culture led to inhibition of deacetylases and to an increase in acetyltransferase activity [179]. At the same time, treatment of MDA-MB-231 and MCF-7 human breast cancer cell cultures resulted in the inhibition of activity of p300 acetyltransferase [180]. Resveratrol can enhance activity of SIRT1 histone deacetylase by binding to its active site, thus leading to the inhibition of the proliferation of breast cancer cells derived from BRCA-1-knockout mice [181]. This plant phenolic compound can also attenuate one of the multiple activities of NuRD histone deacetylase complex by destabilizing its interaction with MTA1 coregulator protein, thus increasing the concentration of active acetylated forms of p53 and PTEN tumor suppressors [182,183]. Treatment with genistein promotes acetylation of histones at the transcription start sites of p16 and p21 tumor suppressor genes in prostate (LNCaP and DuPro) [184] and breast (MDA-MB-231) [185] cancer cell cultures, and of BTG3 tumor suppressor genes in renal (A498, ACHN) [171] and prostate (LNCaP, PC3) [172] cancer cell cultures.

Epigenetic mechanisms can regulate the expression of cellular genes both at the stage of their transcription and by changing the amount or activity of its mRNA. The key components of these regulatory systems are non-coding RNAs (ncRNA) known as microRNAs (miRNAs), capable of inhibiting translation of mRNAs or promoting their degradation [186]. There is a number of other regulatory ncRNAs, such as long or circular ncRNAs, whose regulatory activities towards gene expression are based on different mechanisms [187]. It is well established that aberrant miRNA activity in the cell might promote malignant transformation, as these molecules are involved in regulation of the expression of numerous oncogenes and tumor suppressors [188]. Clinical relevance of non-coding RNAs increases steadily during the last few years; a number of clinical studies consider these regulatory molecules as biomarkers or therapeutic targets [189].

Many plant polyphenols can modulate expression of ncRNA, thus affecting their regulatory activity. It was established that treatment of HepG2 human hepatocellular carcinoma cell culture with EGCG downregulated the expression of 48 different miRNAs, while upregulating expression of another 13 miRNAs at the same time. Upregulated miRNAs included miR-16, which targets mRNA encoding Bcl-2 antiapoptotic protein, causing suppression of the proliferation and induction of apoptosis [190]. Treatment of SK-N-BE2 and IMR-32 malignant human neuroblastoma cell cultures with this phenolic compound suppressed expression of oncogenic miR-92, miR-93, and miR-106b miRNAs, and upregulated expression of tumor suppressors miR-7-1, miR-34a, and miR-99a, causing suppression of the proliferation and induction of apoptosis; treatment with N-(-4-hydroxyphenil)-retinamide potentiated this effect [191]. In CL13 murine lung adenocarcinoma and H1299 human non-small cell lung carcinoma cell cultures, treatment with EGCG suppressed proliferation by upregulating expression of miR-210 caused by the binding of EGCG to hypoxia-induced transcription factor HIF-1α, resulting in its stabilization [192]. This plant phenolic compound can also induce apoptosis in MCF-7 human breast cancer cell culture and mouse xenografts by suppressing the expression of miR-25 [193]. The tumor suppressor miRNAs of the let-7 family is also affected by EGCG: proliferation of human NCI-H446 small-cell cancer and MSTO-211 lung mesothelioma cell cultures was inhibited through the upregulated expression of miRNAs of this family after EGCG treatment [194]. EGCG and quercitin were able to reduce invasiveness of BxPc-3 and MIA-PaCa2 human pancreatic ductal adenocarcinoma cells in culture by upregulating miR-let-7a that belongs to the let-7 miRNA family [195].

Curcumin also modulates miRNA expression. In a recent study, the ability of this polyphenol to induce global changes in mRNA and miRNA expression in MCF-7, MDA-MB-321, and T47D human breast cancer cell cultures was established; the exact pattern of changes in RNA expression was individual for each cell line [196]. Downregulation of oncogenic miR-21 miRNA expression by curcumin suppressed proliferation of human MCF-7 breast cancer [197], AGS gastric adenocarcinoma [198], A549 non-small cell lung adenocarcinoma [199], and Rko and HCT116 colon carcinoma cell cultures [200]. For colon carcinoma cell cultures, a significant decrease in invasiveness and motility was also observed. The ability to inhibit miR-21 expression was demonstrated not only for curcumin, but for its synthetic analogs as well [201,202]. Treatment of MCF-7 human breast cancer cell cultures with this polyphenol downregulated expression of oncogenic miR-19a and miR-19-b, leading to inhibition of proliferation [203]. Curcumin also has the ability to simultaneously modulate the expression of multiple miRNAs: changes in the level of expression of around 30 miRNAs were detected in a BxPC-3 human pancreatic carcinoma cell culture after curcumin treatment [204]. This polyphenol also upregulated four miRNAs and downregulated miR-136 and miR-186* in a A549/DDP human lung adenocarcinoma cell culture, thus leading to induction of apoptosis [205]. The effects of curcumin on expression of miR-34 and miR-98 were also demonstrated: curcumin-mediated increase in the expression of miR-34 led to inhibition of the proliferation of 22RV1, PC-3, and DU145 human prostate carcinoma cell cultures [206] and miR-98 of A549 human lung adenocarcinoma cell culture [207]. In the latter case, treatment with this phenolic compound also decreased invasiveness of tumor cells by downregulating expression of MMP2 and 9 extracellular matrix metalloproteinases.

Modulation of miRNA expression in several cancer cell lines that led to inhibition of the proliferation and induction of apoptosis was also demonstrated for quercitin. Treatment with this polyphenol upregulated expression of miR-16 in HSC-6 and SSC-9 human oral cavity squamous cell carcinoma cell cultures, leading to inhibition of proliferation [208]; similar downregulation of this miRNA expression in A549 human lung adenocarcinoma cell culture by this polyphenol led to a decrease in claudin-2 expression [209]. Suppression of the proliferation and induction of apoptosis by quercitin-induced upregulation of the tumor suppressor miR-34 in HepG2 human hepatocellular carcinoma cell culture was also demonstrated [210]. It was established that treatment with this plant phenolic compound upregulates expression of miR-146a in MCF-7 and MDA-MB-231 human breast cancer cell cultures and mouse xenograft models, leading to inhibition of proliferation [211]. Treatment of BEAS-2B normal human bronchial epithelial cells with quercitin prevented induction of their Cr(VI) ions-induced malignant transformation by downregulation of miR-21 expression, both in cell cultures and in the mouse xenograft model [212]. Quercitin was also able to suppress the proliferative activity of several pancreatic ductal carcinoma cell lines in culture and in xenograft models by upregulating let-7c miRNA expression [213]. In combination with resveratrol, this polyphenol exerted antiproliferative and proapoptotic activities towards HT-29 human colon adenocarcinoma cell culture by downregulating miR-27a [214].

Resveratrol also causes modulatory activity towards miRNA expression in tumor cells. It was demonstrated that treatment of murine C6 glioma cell cultures and xenografts with this phenolic compound downregulated expression of miR-21, miR-30a-5p, and miR19, leading to global changes in cellular proteome and inhibition of proliferative activity [215]. Antiproliferative and proapoptotic activity of resveratrol mediated by downregulation of miR-21 was shown for human U251 glioblastoma [216], T24 and 5637 urinary bladder carcinoma [217], and DU145 prostate carcinoma cell cultures [218]. Treatment of human HT-29 and HCT-116 colon cancer and U87 and U251 malignant glioma cell cultures with resveratrol resulted in a reduction in their invasiveness and motility mediated by upregulation of mir-34c in colon carcinoma and miR-34a in malignant glioma cells [219,220]. In MCF-7 and MDA-MB-231 breast cancer cell cultures, this phenolic compound was able to promote apoptosis by modulating expression of more than 40 miRNAs; miR-542-3p, miR-122-5p, miR-199a-5p, miR-125b-1-3p, miR-140-5p, and miR-20a-5p, known for participation in tumorigenesis, were the most important among them [221].

Genistein also has modulatory activity towards miRNA expression. Treatment of human C918 uveal melanoma C918 [222] and SKOV3 ovarian cancer cell cultures [223] inhibited their proliferation by downregulating miR-27a expression. Remarkably, genistein-induced decrease in the viability of cultivated human A549 non-small cell lung adenocarcinoma cells was mediated by the opposite effect: expression of miR-27a was upregulated by polyphenol treatment [224]. Suppression of the proliferation of human MCF-7 breast cancer cell culture by genistein treatment-induced upregulation of miR-23b expression was demonstrated [225]. This plant polyphenol also downregulated expression of miR-151 in human DU145 and PC3 prostate cancer cell cultures, thus suppressing cellular mobility and invasiveness [226]. Downregulation of miR-155 by genistein has a proapoptotic effect on human MDA-MB-435 aggressively metastasizing breast cancer cell cultures [227]. This phenolic compound also has antiproliferative and proapoptotic effects on human AsPC-1 and MiaPaCa-2 pancreatic cancer [228] and PC3 и DU145 prostate cancer cell lines [229] through upregulation of miR-34a expression. It is noteworthy that in the latter case, HOTAIR long regulatory ncRNA, which is associated with malignant cell proliferation and invasiveness, was one of the molecular targets of miR-34a. During the last few years, a growing number of reports on the activity of plant phenolic compounds towards this class of regulatory ncRNAs appeared [230].

A significant amount of data on the modulatory activity of natural polyphenols and on multiple mechanisms of epigenetic regulation were accumulated to the present day, allowing for the possible development of anticancer and chemopreventive drugs that may utilize this activity to prevent or reverse cancer-associated epigenetic disorders. The list of plant polyphenolic compounds that are known to modulate epigenetic regulation systems is not limited to the compounds described above, and is expanding rapidly [231]. The ability of a single polyphenol to simultaneously affect a number of molecular targets may allow a hypothetical polyphenol-based anticancer drug to affect a broad spectrum of tumors and/or alleviate multiple pre-malignant disorders. At the same time, this potential advantage significantly complicates development and application of such hypothetic drugs, as the effect of a given polyphenol on different epigenetic regulation systems might be highly context-dependent and difficult to predict. The mechanisms behind the effects of plant polyphenols on epigenetic regulatory mechanisms are summarized in Table 2.

## 5. Tumor-Promoting Inflammation

Inflammation is a complex response of innate and acquired components of the immune system after a traumatic or infective injury of tissues that includes activation, recruitment, and proliferation of various immune cells and facilitates tissue regeneration and restoration of its normal functioning. The exact characteristics of a site of inflammation, such as composition of immune cells attracted to it and chemical factors secreted by these cells, are dependent on the nature of the damage that caused the inflammatory response. Acute inflammation usually resolves completely after the inflammation-causing lesion is repaired, but in certain cases, it may switch to a chronic inflammation, or in the case of systemic inflammation, which is not caused by infection or trauma, it might develop as a chronic form. A site of chronic inflammation can be viewed as “a wound that never heals” [232]. Damage repair at the inflammatory site must include elimination of tissue debris, modification of the extracellular matrix, induction of angiogenesis, and survival and proliferation of normal cells. These processes facilitate the return of a damaged tissue to homeostasis, but they may also promote malignant transformation: certain forms of immune response are carried out by the production of DNA-damaging reactive oxygen and nitrogen species by neutrophils and macrophages, the induction of angiogenesis supplies the growing tumor with nutrients and oxygen, and modifications of extracellular matrix facilitate the process of metastasizing [233]. These events mark chronic inflammation as a cancer-enabling characteristic, and anti-inflammatory measures are now considered as an important part of antitumor therapy and chemoprevention.

Transcription factors of the NF-κB family are one of the most important regulators of inflammation-associated gene expression. Each of these factors have the Rel homology domain, which is responsible for dimerization of NF-κB proteins and their binding to DNA [234]. Inactive NF-κB dimers form ternary complexes with IκB inhibitor proteins; ubiquitination and subsequent proteasomal degradation of IκB are necessary steps in NF-κB activation. The canonical pathway of NF-κB activation involves phosphorylation of IKK1/2 IκB kinases mediated by NEMO (NF-κB essential modulator) scaffold protein bound to the kinases. This NF-κB activation pathway can be activated by the signals from toll-like receptors (TLR), interleukin-1 receptors (IL-1R), and tumor necrosis factor receptors (TNFR) [235]. The non-canonical NF-κB activation pathway is independent of NEMO activity and is activated by signals from the B-cell activating factor receptor (BAFFR), lymphotoxin β receptor (LTβR), receptor activator of NF-κB (RANK), TNFR2, and fn14. Activation of these receptors leads to activation of NF-κB-inducing kinase (NIK) that phosphorylates IKK1. Activated IKK1 phosphorylates the p100 NF-κB precursor protein, inducing its processing and thus leading to an increase in the concentration of active NF-κB p52 in cytoplasm [236]. IKK1 also phosphorylates those p100 molecules that are bound to preexisting NF-κB dimers, thus acting as their inhibitors (inhibitory p100 is known as IκBδ). Phosphorylation of IκBδ leads to its dissociation and to activation of the NF-κB dimer it was bound to. Canonical and non-canonical pathways of NF-κB activation differ by the intensity and duration of NF-κB activation and by genes regulated by these pathways: the canonical pathway is mainly responsible for activation of an inflammation-associated response, while the non-canonical pathway is involved in cell differentiation and organogenesis [237]. Activation of NF-κB stimulates the cytotoxic immune cell-mediated antitumor activity [238], and also induces expression of antiapoptotic genes, pro-inflammatory cytokines (TNF-αandIL-1, 6 and 8) [239], angiogenesis factors (VEGFand its receptors) [240], and cell metabolism regulators (HIF-1α) [241], thus facilitating the malignant transformation.

Plant phenolic compounds might inhibit NF-κB activation at different stages of signal transduction (Figure 3). Treatment of HCA-7 human colon adenocarcinoma cell culture with EGCG led to the general decrease in NF-κB activity [242]. This phenolic compound was also able to inhibit IKK activity in A549 human lung cancer cell culture, preventing the activation of NF-κB by TNF-α [243]. EGCG also prevented IL-1β-mediated degradation of IRAK receptor-associated kinase in this cell culture, thus inhibiting IKK activation, IκBα degradation, and NF-κBp65 phosphorylation [244]. Inhibition of IKK activity by this phenolic compound was also demonstrated for rat IEC-6 normal small intestine epithelial cell culture [245]. Treatment of rheumatoid arthritis-associated synovial fibroblast culture with EGCG prevented activation of IKK by inhibiting TAK-1 TNF-β-dependent kinase [246,247]. Molecular modeling studies of EGCG interaction with components of NF-κB signal transduction pathways suggested that the inhibitory activity of this compound may be mediated by its interaction with ATP binding site ofTAK-1 and IRAK kinases. The inhibitory activity of this plant phenolic compound towards NF-κB signaling pathway was also demonstrated in vivo: treatment with EGCG alleviated the development of picrylsulfonic acid-induced colitis in Wistar rats by inhibiting activation of this transcription factor, presumably by binding to the IKK inhibitor binding site, as suggested by the results of molecular docking [248]. The ability of EGCG to inhibit the interaction of NF-κB with DNA by covalent modification of thiol groups of cysteine residues of this protein was also demonstrated [249].

The inhibitory activity of curcumin towards NF-κB signaling pathway was also demonstrated. Treatment of MDA-MB-231 human breast cancer cell cultures with this phenolic compound inhibited NF-κB-mediated expression of CXCL-1 and -2 pro-inflammatory cytokines by preventing IκB phosphorylation [250]. In RAW264.7 murine macrophage cultures, curcumin was able to reduce the efficiency of NF-κB binding to DNA, thus leading to inhibition of TNF-α, IL-1β, and IL-6 expression [251]. Although this polyphenol failed to demonstrate similar activity towards MCF-7 human breast adenocarcinoma cell cultures, curcumin-loaded chitosan-protamine nanoparticles did reduce the levels of NF-κB, IL-6 and TNF-α [252]. Treatment of T47D human breast cancer cell culture with curcumin suppressed the activity of NF-κB signaling pathway both by suppressing activity of this transcription factors and by downregulating expression of NF-κB and IKK [253]. Downregulation of NF-κB expression upon curcumin treatment in HeLa human cervical carcinoma cell culture was also observed [254]. Treatment of Eca109 and EC9706 esophageal squamous cell carcinoma cell cultures and xenografts with this plant polyphenol also resulted in downregulation of NF-κB expression, as well as in suppression of IκB phosphorylation [255]. In WERI-Rb-1 human retinoblastoma cell culture, curcumin suppressed the activation of NF-κB, thus preventing its nuclear translocation and downregulating expression of VEGF and MMP-2 and -9 [256].

Quercitin has activity towards NF-κB and its regulators. Treatment of Caco2 human colon adenocarcinoma cell culture with this phenolic compound downregulated expression of NF-κBp65, thus suppressing cellular motility and invasiveness [257]. Expression of the other components of the NF-κB signaling pathway was also affected by quercitin. In Caco-2 and SW-620 human colon carcinoma cell cultures, this polyphenol demonstrated pro-apoptotic activity by upregulating expression of IκBα, suppressing its phosphorylation, and reduction in the efficiency of NF-κB binding to DNA [258]. It was also established that in H460 non-small cell lung cancer cell cultures, quercitin was able to promote apoptosis by downregulating expression of NF-κB while simultaneously upregulating expression of IκBα [259]. Treatment of SAS human oral squamous cell carcinoma cell culture with this phenolic compound resulted in suppression of cellular invasiveness by simultaneously reducing both NF-κB and IκB expression and lowering the presence of phosphorylated IKK1/2 [260]. Quercitin also exerted its activity towards NF-κB during in vivo studies: treatment with this compound abrogated induction of oral adenocarcinoma in Syrian hamster buccal pouches with dimethylbenzanthracene by promoting apoptosis of tumor cells and downregulating expression of NF-κBp50 and p65 [261]. Similar results were obtained during induction of tumorigenesis by dimethylhydrazine in rats: treatment with quercitin lowered the number of colonic tumors and prevented the upregulation of NF-κB expression [262].

A certain degree of controversy exists regarding the modulatory activity of resveratrol towards the NF-κB signaling pathway. It was established that treatment of several human cervical cancer cell cultures with this phenolic compound resulted in reduction in the proliferation and induction of apoptosis; for CaSki cell cultures, these effects were the most prominent. In C33A, HeLa, CaLo, and CaSki cell cultures, the antiproliferative and proapoptotic activities of resveratrol were accompanied by a significant decrease in NF-κB p65 expression [263]. In human colon cancer cell cultures, resveratrol prevented nuclear translocation of NF-κB, presumably by binding to its monomer, and thus preventing its dimerization [264]. This phenolic compound was also able to suppress proliferation of human MV3 and A375 melanoma cells, both in culture and in a mouse xenograft model, by downregulating the expression of NF-κB and NF-κB-regulated miR-221, thus leading to the enhancement of the expression of TFG tumor suppressors, the mRNA of which is targeted by miR-221 [265]. The chemopreventive activity of resveratrol, mediated by inhibition of NF-κB activity with this polyphenol, was also reported. Treatment of KC mice that spontaneously develop pancreatic dysplasias and malignant tumors with resveratrol led to reduction in the number and severity of pre-malignant lesions and to suppression of NF-κB activity in these structures [266]. After induction of hepatocelular carcinoma in rats by ethanol and aflatoxin treatment, this plant phenolic compound alleviated the induction of malignant transformation by preventing the reduction in the activity of antioxidant enzymes and increasing the NF-κB-inhibiting SIRT1 activity [267]. Nevertheless, treatment with resveratrol did not result in suppression of tumor cell growth. This polyphenol was able to suppress proliferation of NF-κB-overexpressing SKOV3 human ovarian cancer cells in aggregates by inhibiting activation of this transcription factor. At the same time, resveratrol failed to demonstrate any significant antiproliferative activity towards OVCAR5 ovarian cancer cell aggregates, whose expression of NF-κB is far less pronounced [268]. Treatment with resveratrol also had no effect on the secretion of VEGF by OVCAR5 cells, while treatment of SKOV3 aggregates with this phenolic compound resulted in a significant decrease in the efficiency of the secretion of this factor. Moreover, secretion of pro-survival interleukin IL-8 by both cell cultures increased significantly upon treatment with resveratrol. Growth stimulatory activity of low concentrations (less than 10 µM) of resveratrol towards MDA-MB-495c human breast cancer cell culture was reported; at the same time, this polyphenol has antiproliferative activity towards MDA-MB-231 and MCF-7 human breast cancer and DU145 human prostate adenocarcinoma cell cultures in all concentration values [269]. The stimulatory effect of resveratrol treatment on MDA-MB-495c cell culture was exerted by promotion of IκB phosphorylation and NF-κB expression, and by facilitation of nuclear translocation of NF-κB.

Certain cases of genistein activity towards the NF-κB signaling pathway are also controversial. Genistein causes pro-apoptotic activity towards U266 human multiple myeloma cell culture by upregulating miR-29b that suppresses NF-κB p65 expression [270]. It was established that treatment of CAL-62 and CGTH-W1 human thyroid carcinoma cell cultures with this phenolic compound resulted in a decrease in the viability of cultured cells through a reduction in the amount of mRNAs of several pro-tumorigenic proteins, including NF-κB [271]. Reduction in the amount of NF-κB mRNA after genistein treatment was also detected for A549 human lung carcinoma, along with pro-apoptotic activity and downregulation of AKT, HIF1, and COX-2 expression [272]. In HT-29 human colon carcinoma cell culture, this polyphenol promoted apoptosis and reverted the epithelial-to-mesenchymal phenotype transition by downregulating expression of NF-κB and Notch-1, thus causing a decrease in expression of invasiveness-related proteins and upregulation of the expression of pro-apoptotic factors [273]. Results of treatment of MDA-MB-231 human breast cancer cell culture with genistein also suggest involvement of Notch-1 in the inhibition of the NF-κB signaling pathway by this phenolic compound, as genistein exerted antiproliferative and proapoptotic activity towards this cell culture by downregulating expression of Notch-1, accompanied by inhibition of NF-κB activation [274]. Genistein was also able to suppress proliferation of LoVo and HT-29 human colon carcinoma cell cultures by preventing IκBα phosphorylation, thus abrogating phosphorylation of NF-κB p65 and its nuclear translocation [275]. It is noteworthy that genistein- and daidzein-rich biotransformed soybean extract has pro-apoptotic activity towards A375 human melanoma cell culture by the opposite mechanism: overexpression of TNF receptors TNFR1/2 caused by this treatment led to IKK activation and to an increase in NF-κBp65 phosphorylation [276]. Moreover, treatment of HT-29 and SW620 human colon adenocarcinoma cell cultures with genistein decreased the viability of the latter culture [277]. This decrease in viability, however, was induced by an increase in H_2_O_2_ concentration caused by the upregulation of the expression of antioxidant enzymes, such as SOD1 and 2, accompanied by activation of NF-κB nuclear translocation and subsequent stimulation of the expression of pro-inflammatory proteins, such as TNF and CXC chemokines. These effects were observed for both cell cultures, but in HT-29 cell culture, they had no significant proapoptotic effect.

The anticancer and chemopreventive activity of plant polyphenols, carried out through activation of the NF-κB signaling pathway by these compounds, was demonstrated on a multitude of objects, including cell cultures, tumor xenografts, and the models of induced carcinogenesis. Suppression of NF-κB activation led to a decrease in the efficiency of antiapoptotic, pro-angiogenic, and prometastatic inflammatory signals in tumor tissue. Mechanisms of modulation of the NF-κB signaling pathway by plant polyphenols are summarized in Table 3. The anti-inflammatory activity of natural phenolic compounds towards normal tissues was also demonstrated, and the mechanisms underlying this activity were not limited to inhibition of NF-κB activation [278]. These mechanisms may also include inhibition of activity of the other key pro-inflammatory enzymes and signaling molecules, such as COX-2 cyclooxygenase [279], iNOS inducible NO synthase^275^, and pro-inflammatory cytokines [280]. These properties suggest the possible use of polyphenols as chemopreventive drugs that suppress pro-tumorigenic influence of inflammation during the entire tumorigenesis. Nevertheless, evidence of growth-stimulatory and pro-inflammatory activity of natural phenolic compounds in cell cultures calls out for the additional study of the activity of polyphenols in the inflammation sites before such drugs could be designed and introduced in clinical practice.

## 6. Interactions with Cancer-Associated Microbiota

Human organisms are inhabited by microorganisms that modulate normal physiological processes [281] and various diseases [282], including malignant tumors [283]. Certain microorganisms of human microbiota might produce genotoxic substances capable of damaging the host cellular genome; colibactin-producing *Escherichia coli* [284] and cytolethal distending toxin-producing *Campylobacter jejuni* [285] are the examples of such microorganisms. Other bacterial toxins may promote tumorigenesis by modulation of cellular signaling pathways: toxins produced by *F. nucleatum* activate the β-catenin signaling pathway and suppress the antitumor activity of NK cells [286]. CagA and VacA proteins secreted by *H. pylori* attenuate activity of tumor suppressors, stimulate proliferation, facilitate the acquisition of the mesenchymal phenotype, and modulate the host immune response [287]. Accumulated evidence of procarcinogenic activity of this microorganism was sufficient to classify it as a IARC class 1 carcinogen [288]. Non-specific products of bacterial metabolism might also affect the process of tumorigenesis. Bacterial metabolites of fatty acids, trimethylamine-n-oxide, and hydrogen sulfide facilitate development of tumors, whereas short-chain fatty acids (SCFA) and niacin suppress this process [289]. It is also established that colonic malignancies are often accompanied by changes in the species composition of gut microbiota [290], but it is largely unclear whether these changes are the cause of tumorigenesis or the consequence of this process [291].

Plant polyphenols have antibacterial properties [292], which allow them to modulate the intestinal microflora. Silibinin was able to suppress the *H. pylori* culture growth by promoting changes in bacterial morphology and suppressing cell division [293]. Kaempferol and (-)-epicatechin demonstrated antibacterial and bacteriostatic effects towards this pathogen in vitro [294]. Similar activity was demonstrated for ellagic and gallic acid, as well as for quercitin aglycon and its glycosylated form [295]. Antibacterial activity of baicalin and baicalein towards this microorganism was registered both in vitro and in vivo: these phenolic substances decreased the bacterial load in *H. pylori*-infected mice by inhibiting the adhesiveness and invasiveness of this microorganism [296]. Antibacterial activities of polyphenol-containing natural and artificial mixtures were also detected. Citrus juices and extracts from citrus plants have antibacterial activity towards *H. pylori* culture in vitro, and alleviated *H. pylori*-induced gastritis in vivo by suppressing bacterial colonization of mucosa and abrogating inflammation elicited by the infection [297]. Similar antibacterial activity of extracts of fruits of *Rubus crataegifolius* and bark of *Ulmus macrocarpa* was described [298]. The antibacterial activity of plant polyphenols towards *H. pylori* was determined by the ability of these substances to impair functioning of transmembrane energetic of this microorganism [299].

Natural phenolic compounds demonstrated antibacterial properties towards *C. jejuni* as well. The antibacterial activity of curcumin was demonstrated by treatment of *C. jejuni* culture with this substance [300]. Similar antibacterial properties of EGCG were also demonstrated, along with the ability of this polyphenol to impair motility and biofilm formation by these microorganisms and to suppress activity of bacterial autoinducer-2, thus disturbing quorum-sensing activity [301]. Treatment with resveratrol prevents *C. jejuni*-induced loosening of cellular junctions between intestinal epithelial cells and alleviates infection-induced apoptosis both in vitro and in vivo, thus facilitating epithelial barrier function and preventing development of inflammation [302,303].

EGCG, theaflavines and other tea polyphenols have antibacterial activity towards *F. nucleatum* in vitro by disturbing integrity of the plasma membrane of this organism and by reducing its adhesiveness along with the ability of biofilm formation [304]. Similar activity was demonstrated for complexes of resveratrol with cyclodextrane together with inhibitory activity towards *F. nucleatum*-induced inflammation [305]. Treatment of female rats on a high-fat diet with genistein prevented the increase in the presence of *Enterobacteriaceae* in the microflora of their progeny, accompanied by a lower risk of mammary cancer recurrence [306]. The antibacterial activity of plant polyphenols towards cancer-associated microorganisms is summarized in Table 4.

Microbiota-associated chemopreventive activity of plant polyphenols may be carried out not only by their antibacterial activity towards certain microorganisms, but also by modulation of species composition of gut microbiota. Dietary supplementation of sausages with powder of dried anthocyanine-rich berries reduced the number of intestinal tumors induced by azoxymethane treatment in rats, and alleviated the presence of pro-inflammatory *Bilophila wadsworthia* in their microflora [307]. Treatment of IL10^−/−^ mice with curcumin in a similar tumorigenesis induction experiment resulted in a decrease in tumor burden and in prevention of cancer-associated changes in microbiota [308]. Anthocyanine-rich extract from *Rubus occidentalis* berries was able to reduce the number of tumors and abrogate inflammatory response during azoxymethane-induced carcinogenesis in mice, along with promotion of *Neisseria* and butyrate-producing bacteria growth and suppression of pathogenic *H. pylori*, *Campylobacter*, *Bacteroides,* and *Prevotella* [309]. Treatment of rats with polymethoxyflavone mix during induction of carcinogenesis by benzo-a-pyrene prevented changes in species composition of intestinal microbiota by alleviating the increase in the abundance of *Sphingobacteriia*, *Bacilli* and *Gammaproteobacteria* classes, *Erysipelotrichales* and *Lactobacillales* orders, and the *Parabacteroides* family, and by promoting an increase in the abundance of butyrate-producing *Ruminococcaceae* family [310]. Treatment of rats with isoliquiritigenin during azoxymethane/dextransulfate-induced carcinogenesis resulted in the reduction in tumor incidence by preventing the increase in the abundance of *Firmicutes* phylum in their microflora, while simultaneously decreasing abundance of *Bacteroidetes* [311]. This phenolic compound also decreased the presence of opportunistic pathogens, such as *Escherichia* and *Enterococcus*, while simultaneously increasing the abundance of butyrate-producing bacteria, such as *Bytiricicoccus*, *Ruminococcus,* and *Clostridium*. Induction of carcinogenesis with a high-fat diet in APC*^min/+^* mice was abrogated by neohesperidin by inducing the opposite direction of changes of abundance of *Firmicutes* and *Bacteroidetes*: this phenolic compound decreased the abundance of *Bacteroidetes* and increased the abundance of *Firmicutes* and *Proteobacteria* [312]. The opposite direction of changes in the relative abundance of *Firmicutes* and *Bacteroidetes* in these experiments might suggest that the analysis of cancer-associated microbiota changes at the phylum level may not be informative enough, as only a portion of species of aforementioned phyla are associated with carcinogenesis [313]. Results of the experiments of cancer chemoprevention in azoxymethane-treated mice with EGCG speak in favor of this hypothesis. Treatment with EGCG reduced the severity and number of tumors by preventing the changes in the abundance of different taxa of the *Firmicutes* phylum: this plant phenolic compound prevented the increase in the abundance of *Anaerotruncus*, *Faecalibacterium* and *Streptococcus* and decrease in the abundance of *Clostridiaceae*, *Lactobacillis* and *Lachnospiraceae*^311^ [314]. EGCG also promoted an increase in the abundance of *Bacteroides* and decrease in the abundance of *Fusobacterium*, *Ruminococcus,* and *Veillonella*. Dietary supplementation of mice xenograft with MCA-205 and E0771 murine cancer cells with castalagin-rich *Myrciaria dubia* berries led to sensibilization of MCA-205 to PD-1 inhibitor therapy, and circumvention of resistance of E0771 to these inhibitors [315]. Castalagin exerted this potentiating effect by promoting the selective enrichment of murine gut microbiota with *Alistipes*, *Blautia,* and *Ruminococcus*.

These data emphasize the interplay between intestinal microbiota, the host organism immune system, and tumor microenvironment, and outline its crucial role in the process of tumorigenesis. Contacts with bacteria and their presence in the human organism since the earliest postnatal period are essential for priming the immune system for response to pathogen infections [316]. The interaction of bacteria and their metabolites with host dendritic cells is necessary for developing different immune reactions to commensal vs. pathogenic microorganisms [317]. Activity of intestinal mucosa-associated T lymphocytes is also modulated by microbiota: presence of microorganisms facilitate development CD4+ and cytotoxic CD8+ T lymphocytes [318]. Induction of peripheral RORγ+ regulatory T lymphocytes (Tregs) is responsible for regulating the inflammatory response and maintaining homeostasis of microbiota and is also dependent on intestinal microbiome composition during the early period of life [319,320]. Bacterial metabolites are also involved in regulation of the effect of the immune system on intestinal mucosa. Short-chain fatty acids decrease abundance of Th1/Th17 T helpers while increasing the presence of Tregs, thus reducing the intensity of inflammatory response [321,322]. Secondary bile acids were also able to modulate differentiation of Th17 and RORγ+ Tregs [323,324]. The relative abundance of different components of the immune system plays a crucial role in the determination of the outcome of immune response: an increase in abundance of tumor-associated macrophages and fibroblasts, Tregs, regulatory B cells, and myeloid-derived suppressor cells (MDSCs) is associated with rapid and unimpeded tumor growth [325], while increase in the abundance of CD8+ cytotoxic T lymphocytes, natural killer (NK) cells, and M1 macrophages and CD4+ T lymphocytes is associated with the direct destruction of tumor cells or with stimulation of this process [326,327].

Plant polyphenolic compounds can modulate tumor microenvironments either indirectly, by regulating the composition of intestinal microbiota, or directly, by mediating interactions between different types of immune cells. For example, polyphenols can stabilize the relative abundance of different immune cells on the intestinal mucosa and prevent development of chronic inflammation by preventing carcinogen-induced dysbiosis. Moreover, maintenance of intestinal eubiosis can be associated with sensitivity to immune checkpoint therapy; however, the exact molecular mechanisms responsible for this association are yet to be determined [328].

Mechanisms of the modulation of tumor immune microenvironments by plant phenolic compounds are not limited to prevention of dysbiosis. During multiple in vitro experiments, an increase in the abundance of CD8+ T lymphocytes and enhancement of their cytotoxic activity upon polyphenol treatment were observed [329]. Thus, treatment of the murine Lewis lung adenocarcinoma model with curcumin led to suppression of MDSCs, resulting in an increase in the number of CD8+ T lymphocytes [330]. Cytotoxic activity of NK cells can also be increased by natural phenolic compounds: treatment of human L3.4 pancreatic ductal adenocarcinoma cell culture and MIA PaCa-2 pancreatic adenocarcinoma cell culture with curcuminoids led to the enhancement of cytotoxic activity of co-cultured donor NK cells [331]. Immunomodulatory effects of plant phenolic compounds that increase the anti-tumor immune response were also observed during in vivo studies. Treatment of mice xenografted with B16F10 murine melanoma cells with resveratrol led to the increase in intra-tumoral NK cell cytotoxicity; stimulation with IL-2 additionally enhanced this effect [332]. Life-long intake of genistein by rats resulted in lower incidence of mammary tumors upon stimulation of tumorigenesis by dimethylbenzanthracene, as well as the suppression of CD4+ Tregs proliferation and an increase in the abundance of CD8+ cytotoxic T lymphocytes, thus resulting in enhancement of the sensitivity of these tumors to tamoxifen [333].

Interactions between immune cells in tumor microenvironments are targeted by the immune checkpoint inhibitors—anticancer drugs that prevent the suppression of cytotoxic T lymphocyte activity [334]. These drugs can be used in combination with other types of anticancer therapy, such as photodynamic therapy [335]. Among several different immune checkpoints, the system that consists of PD-1 receptor and PD-L1 ligand system is the most studied to date. Activation of membrane PD-1 of T lymphocytes leads to a decrease in their activity, proliferation, and survival [336]. Recent studies indicate that plant phenolic compounds can modulate this regulatory mechanism. Thus, resveratrol was able to attenuate its activity by preventing PD-L1 glycosylation and dimerization in JIMT-1 human breast adenocarcinoma cells that overexpress this ligand [337]. Treatment with apigenin prevented induction of PD-L1 by γ-interferon in several melanoma cell cultures in vitro, and enhanced infiltration of T lymphocytes in murine xenografts of these melanoma cells, while simultaneously suppressing expression of PD-L1 in dendritic cells [338]. The similar activity of apigenin, along with luteolin, towards KRAS-mutated non-small cell lung carcinoma cell cultures and xenografts was described: these compounds were able to prevent γ-interferon-induced upregulation of PD-L1 expression by suppressing activity of the MUC-1C/STAT3 signaling pathway; molecular docking results suggest that these polyphenols can interact directly with STAT3 molecules [339]. Curcumin was also able to downregulate PD-L1 expression in Cal27 and FaDu oral squamous cell carcinoma cultures, as well as in the murine model of induction of oral squamous cell carcinoma by 4-nitroquinoline oxide [340]. This phenolic compound also demonstrated similar PD-L1 downregulation in Hep3B and CSQT-2 hepatocellular carcinoma cell cultures, an exerted synergistic effect with the anti-PD-1 therapy of Hep3B murine xenografts [341]. Several other polyphenolic compounds have inhibitory activity towards the PD-1/PD-L1 immune checkpoint; EGCG, hesperidin, baicalin, and quercitin are the examples of these polyphenols [342]. At the same time, it is important to emphasize that plant phenolic compounds can also demonstrate the opposite activity: thus, treatment of A549, H460, and H1299 lung adenocarcinoma cell cultures with resveratrol led to an increase in Snail protein stability, which in turn led to the stimulation of the Wnt signaling pathway, and ultimately, upregulated the PD-L1 expression [343]. These data point out the necessity of further investigation of PD-1/PD-L1 immune checkpoint functioning, and of the mechanisms of plant polyphenol-mediated changes in this regulatory system.

When considering the antitumor and chemopreventive activity of plant polyphenols, another important aspect should be taken into account: natural phenolic compounds are not metabolically inert, and may undergo extensive chemical modification by host microbiota [344]. It is established that the products of the bacterial metabolism of polyphenols might have different activities as compared to the original compounds [345]. The most prominent examples of such metabolites are equol [346,347] and urolithins [348,349], products of the modification of daidzein and ellagitannins, respectively. Their antitumor and chemopreventive activity was demonstrated in a number of studies.

Treatment of MCF-7 human beast adenocarcinoma cell culture resulted in the suppression of the proliferation and induction of apoptosis by upregulating miR-10a-5p expression and subsequent inhibition of the PI3K/AKT signaling pathway [350]. In HCT-15 colon cancer cell cultures, equol was able to enhance expression of Nrf2, thus suppressing proliferation of this culture [351]. Equol also exerted antimetastatic activity: treatment of MDA-MB-231 human breast cancer cell culture with this compound led to a decrease in the invasiveness of these cells by inhibiting MMP-2 expression [352]. Treatment of MCF10 human breast cancer cell culture with equol led to abrogation of TNF-1-mediated activation of Gli1, a member of the hedgehog signaling pathway, which is responsible for induction of cellular motility [353]. The chemopreventive activity of equol was also described: dietary supplementation of rats with this compound alleviated the efficiency of dimethylbenzanthracene-induced carcinogenesis [354]. Similar dietary supplementation reduced the efficiency of carcinogenesis by urethane in mice, while simultaneously increasing the level of superoxide dismutase and reducing the concentration of oxidative stress markers in blood serum [355].

The anticancer activity of urolithins was also demonstrated: treatment of Caco-2 colon cancer cell culture with these compounds resulted in suppression of the proliferation and induction of apoptosis [356], likely caused by the upregulation of CDKN1A expression [357]. In 22RV1 and LNCaP human prostate cancer cell cultures, various urolithins were able to induce apoptosis and to increase p53 expression [358]. The similar activity of these substances towards the expression of p53 in colon cancer cell cultures was described [359]. Treatment of these cultures with urolithin also led to cell cycle arrest, apoptosis induction, and an increase in ROS production. Interestingly, treatment of normal human fibroblast cell cultures with this phenolic compound had the opposite effect on ROS production [360]. Dependence of the direction of oxidative stress modulation by urolithins on the type of cell culture was demonstrated in a number of studies, and the mechanisms behind this phenomenon include the condition of cellular antioxidants and oxidative stress-inducing factors [361]. The proapoptotic and antiproliferative activity of urolithins is not limited to the mechanisms mentioned above: these substances might also modulate PI3K/AKT, NF-κB, and WNT/β-catenin signaling pathways [349].

## 7. Conclusions

Data accumulated over the decades of study on the effect of natural polyphenols on human health suggest that these compounds exert antitumor and chemopreventive activities by multiple different mechanisms. Plant phenolic compounds can suppress metabolic activation of procarcinogens and reduce the levels of highly reactive compounds in the cell, thus preventing damage of genomic DNA. These substances demonstrate activity towards factors involved in epigenetic regulation, such as miRNAs and DNA- and histone-modifying enzymes, the misregulation of which probably contributes to malignant transformation. The ability of polyphenols to mitigate the pro-survival and growth-stimulatory effects of pro-inflammatory signals on tumor cells was also demonstrated. It was also established that phenolic compounds can modulate the composition of the intestinal microbiota throughout the entire process of tumorigenesis by suppressing growth of particular bacterial species and by changing the overall abundance of different taxa.

At the same time, it was also established that the direction of the effect of a given natural phenolic compound on cellular metabolism depends on a multitude of factors, such as the exact chemical structure of a polyphenolic molecule and the type of cell culture or parameters of an animal model. In some cases, the effects of different plant polyphenols on a given physiological process were opposite: natural phenolic molecules might act either as AhR antagonists, thus suppressing expression of phase I xenobiotic metabolism enzymes, or as agonists of the same receptor. Under different conditions, polyphenols demonstrated both prooxidant and antioxidant activities. Increasing pro-inflammatory signaling by plant polyphenols was also described. The ambiguous nature of the influence of polyphenols on physiological processes involved in malignant transformation raises doubts about the possible anticancer and chemopreventive applications of these compounds. To rule out these doubts, an in-depth understanding of the mechanisms of interaction of phenolic substances with biological macromolecules and the physiological consequences of these interactions is required. One of the possible ways to achieve this degree of understanding is the application of high-throughput omics approaches to the study of polyphenol activity [362].

High levels of chemical modifications of natural phenolic compounds by host microbiota, along with poor solubility of these compounds resulting in a low bioavailability, pose another challenge to their therapeutic application. This problem affects both purified polyphenols and polyphenol-rich foods and extracts. In some cases, even high doses of phenolic substances supplemented to individuals did not result in any significant presence of the substance in blood serum [363]. To circumvent the poor bioavailability of natural phenolic compounds, new formulations are currently being developed, such as various nanoparticles and liposomes [364].

Despite the long history of extensive study and the multifaceted antitumor and chemopreventve activity that encompass a vast number of physiological processes, it is still too early to introduce the polyphenol-based drugs into the clinical practice as tertiary chemopreventors. Yet, a polyphenol-rich diet could already be considered as an effective primary and secondary chemopreventive measure due to the multi-targeted blocking and suppressing activity exerted by the phenolic compounds it contains.

## Figures and Tables

**Figure 1 ijms-24-10663-f001:**
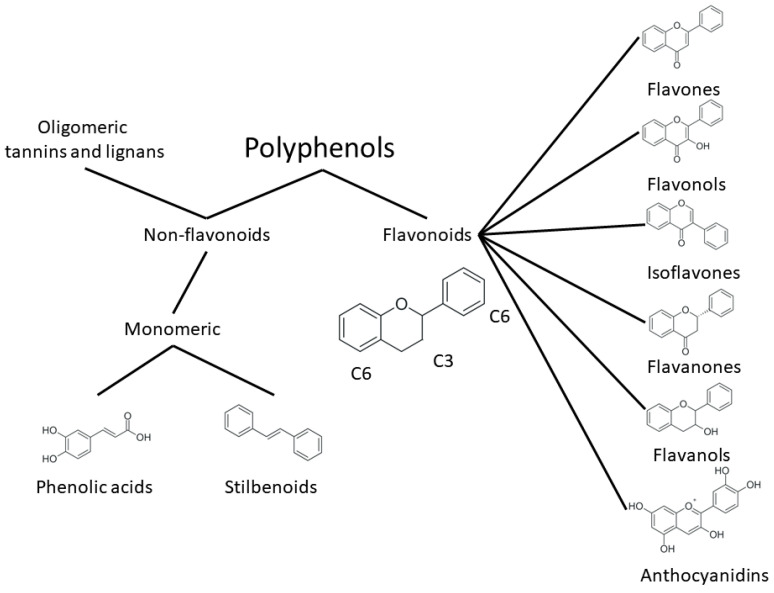
Structure and classification of plant phenolic compounds.

**Figure 2 ijms-24-10663-f002:**
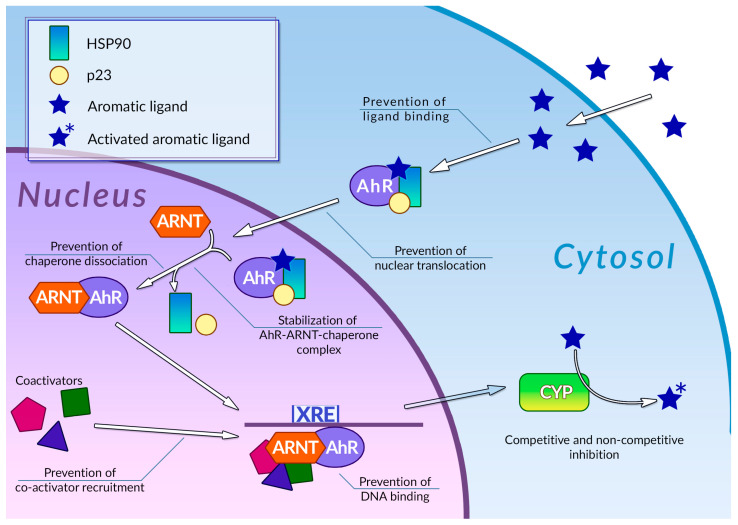
Regulation of phase 1 xenobiotic metabolism and its inhibition by plant polyphenols. See text for detail.

**Figure 3 ijms-24-10663-f003:**
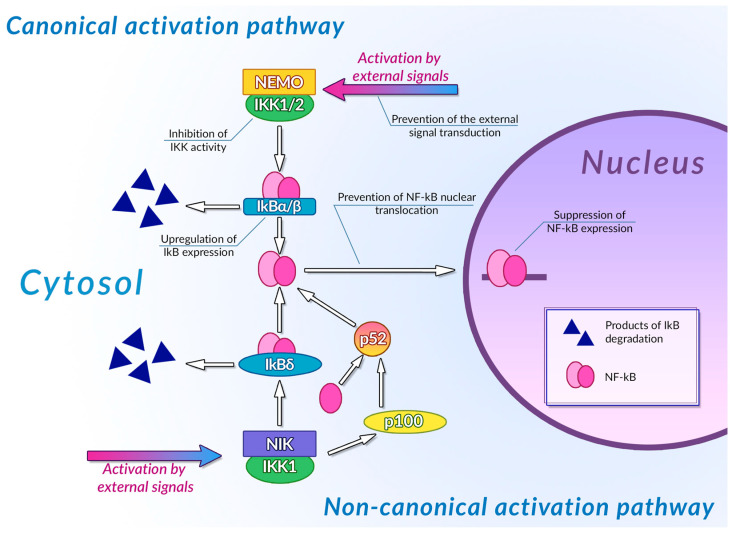
Inhibition of NF-κB signaling pathways by plant polyphenols. See text for detail.

**Table 1 ijms-24-10663-t001:** Mechanisms of action of plant polyphenols on the AhR signal transduction pathway.

Compound	Structure	Mechanism	Reference
Curcumin	Phenolic acid homodimer	Direct competitive inhibition of AhR in cell-free extracts	[92]
Quercetin	Flavonol		[93]
Resveratrol	Stilbenoid		[94]
Flavone	Flavone		[95]
Luteolin	Flavone		
Apigenin	Flavone		
Galangin	Flavonol		
Kaempferol	Flavonol	Inhibition of AhR in Caco2 cell culture	[96]
Luteolin	Flavone		
Apigenin	Flavone		
Fisetin	Flavonol		
Quercetin	Flavonol	Agonism of AhR in Caco2 cell culture	
Robinetin	Flavone		
Taxifolin	Flavonol		
Morin	Flavone		
Quercetin	Flavonol	Stimulation of AhR-regulated luciferase gene expression in H1L6.1c2 reporter cell culture	[97]
Naringin	Glycosylated flavanone		
Hesperidin	Glycosylated flavanone		
Hesperitin	Flavanone		
Galangin	Flavonol	Prevention of nuclear translocation of AhR in MCF-7 and Hepa-1c1c7 cell cultures	[100]
Apigenin	Flavone		
Luteolin	Flavone		
Flavone	Flavone		
Kaempferol	Flavonol		
Curcumin	Phenolic acid homodimer		[101]
Resveratrol	Stilbenoid		[94]
Naringenin	Flavanone	Prevention of active AhR-Arnt heterodimer formation in cell cultures	[100]
Apigenin	Flavone		
Kaempferol	Flavonol		
Galangin	Flavonol		[102]
EGCG	Flavanol and phenolic acid heterodimer		[100,103]
Curcumin	Phenolic acid homodimer	Prevention of XRE binding by AhR-Arnt dimer in tumor cell cultures	[101]
Galangin	Flavonol		[102]
Resveratrol	Stilbenoid		[104]
Quercetin	Flavonol		[93]
Kaempferol	Flavonol		
Resveratrol	Stilbenoid	Prevention of AhR-Arnt coactivator recruitment	[94]
Genistein	Isoflavone		[105]
Kaempferol	Flavonol		[106]

**Table 2 ijms-24-10663-t002:** Mechanisms behind the effects of plant polyphenols on epigenetic regulatory mechanisms.

Compound	Structure	Mechanism	Reference
Curcumin	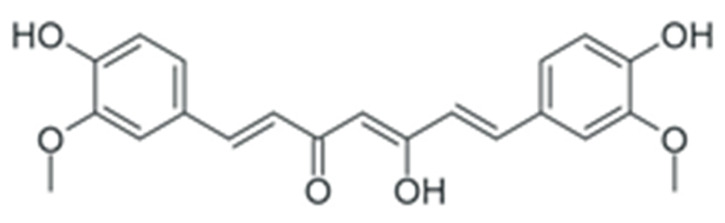	Downregulation of DNMT DNA methyltransferases expression	[164,165]
		Downregulation of histone deacetylase expression	[177,178]
		Modulation of various miRNAs expression	[196]
		Downregulation of miR-21 expression	[197,198,199,200,201,202]
		Downregulation of miR-19a and -19b	[203]
		Modulation of various miRNAs expression	[204]
		Downregulation of miR-136 and -186*	[205]
		Upregulation of miR-34	[206]
		Upregulation of miR-98	[207]
EGCG	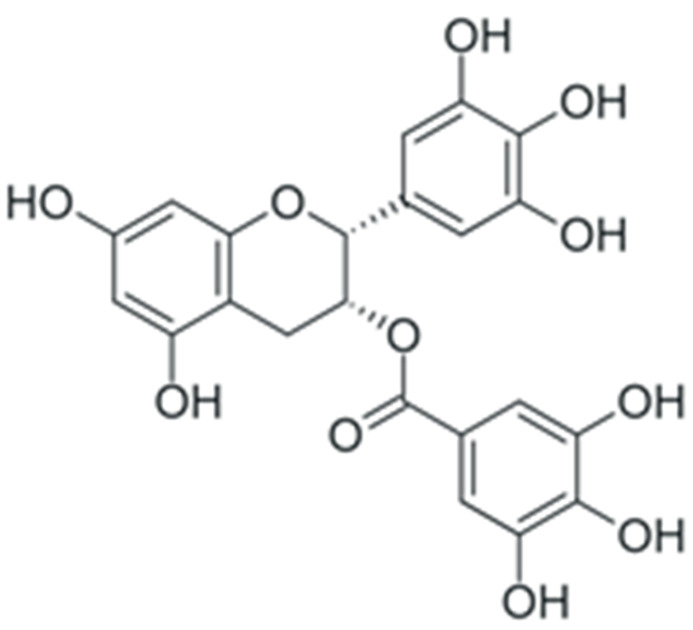	Downregulation of DNMT DNA methyltransferases expression	[160]
		Inhibition of DNMT catalytic activity	[161]
		Inhibition of HDAC 1-3 histone deacetylase expression	[174,175]
		Inhibition of histone acetyltransferase catalytic activity	[176]
		Modulation of various miRNAs expression, including miR-16	[190]
		Downregulation of miR-92, -93 and -106b, upregulation of miR-7-1, -34a and -99a	[191]
		Upregulation of miR-210 expression	[192]
		Downregulation of miR-25 expression	[193]
		Upregulation of let-7 family miRNAs expression	[194]
		Upregulation of miR-let7a (in combination with quercetin)	[195]
Quercetin	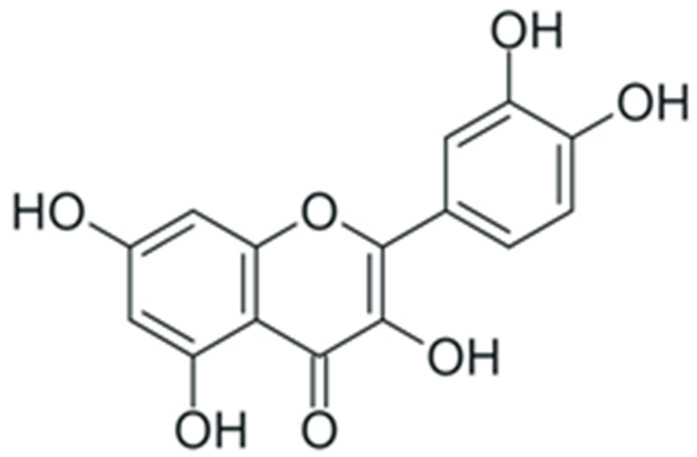	Inhibition of DNMT catalytic activity	[167]
		Inhibition of histone deacetylase and increase in histone acetyltransferase activity	[179]
		Inhibition of p300 acetyltransferase activity	[180]
		Upregulation of miR-16	[208]
		Downregulation of miR-16	[209]
		Upregulation of miR-34	[210]
		Upregulation of miR-146a	[211]
		Downregulation of miR-21	[212]
		Upregulation of let-7c	[213]
		Downregulation of miR-27a (in combination with resveratrol)	[214]
Resveratrol	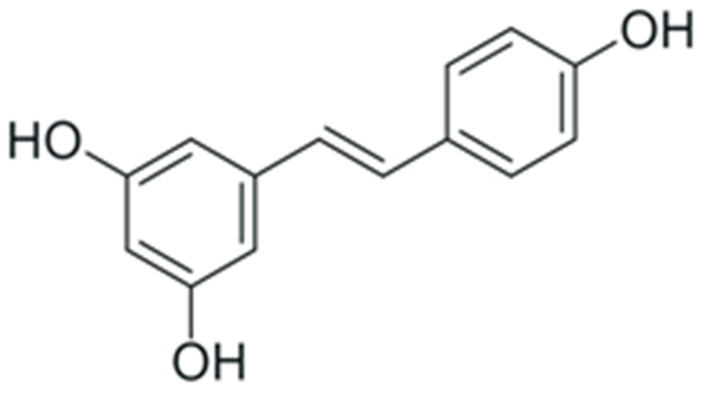	Inhibition of DNA methyltransferase recruitment by SIRT1	[168]
		Selective reversion of hypermethylation of tumor suppressor genes (in combination with pterostilbene)	[169,170]
		Recruitment of DNMT3b DNA methyltransferase to oncogenes	[169]
		Enhancement of SIRT1 histone deacetylase activity	[181]
		Modulation of NuRD histone deacetylase complex activity	[182,183]
		Downregulation of miR-19, -21 and -30-a5p	[215]
		Downregulation of miR-21	[216,217,218]
		Upregulation of miR-34a and -34c	[219,220]
		Modulation of various miRNAs expression, including miR-20a-5p, -140-5p, -125b-1-3p, -199a-5p, -122-5p and -542-3p	[221]
Genistein	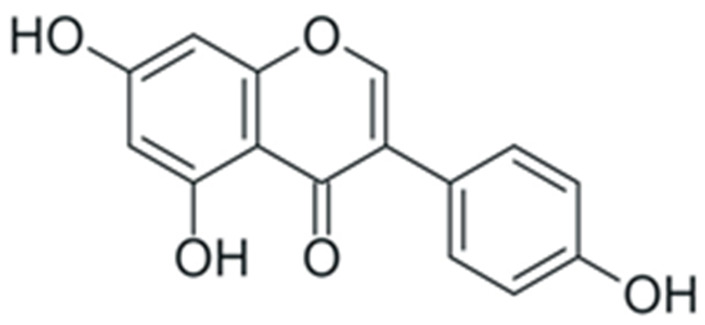	Inhibition of DNMT catalytic activity	[171,172,173]
		Promotion of histone acetylation at regulatory regions of tumor suppressor genes	[171,172,184,185]
		Downregulation of miR-27a	[222,223]
		Upregulation of miR-27a	[224]
		Upregulation of miR-23b	[225]
		Downregulation of miR-151	[226]
		Downregulation of miR-155	[227]
		Upregulation of miR-34a	[228,229]

**Table 3 ijms-24-10663-t003:** Modulation of NF-κB signal transduction pathway by plant polyphenols.

Compound	Structure	Mechanism	Reference
Curcumin	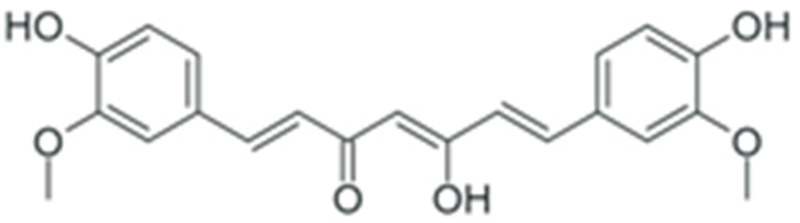	Prevention of IκB phosphorylation	[250]
		Reduction in NF-κB DNA binding efficiency	[251]
		Reduction in NF-κB level	[252]
		Downregulation of NF-κB and IKK expression	[253]
		Downregulation of NF-κB expression	[254]
		Downregulation of NF-κB and suppression of IκB phosphorylation	[255]
		Prevention of NF-κB activation and nuclear translocation	[256]
EGCG	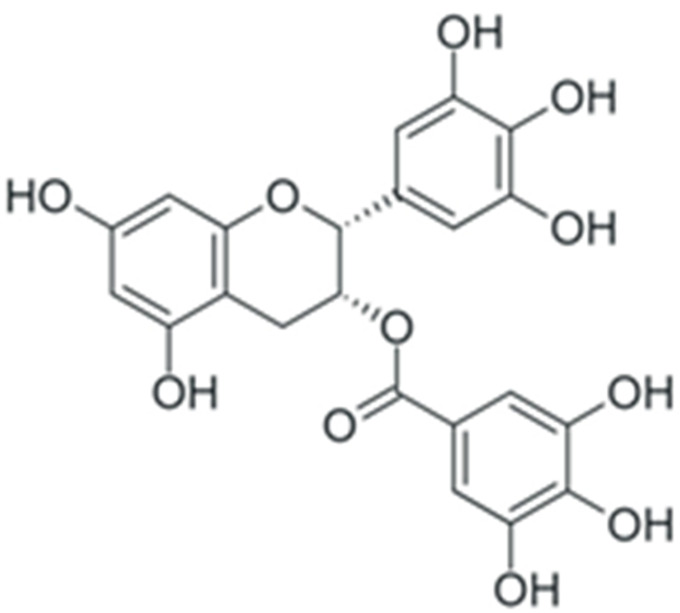	General decrease in NF-κB activity	[242]
		Inhibition of IKK activity	[243,245]
		Prevention of IRAK receptor-asociated kinase degradation	[244]
		Inhibition of TAK-1 TNF-β-dependent kinase	[246,247]
		Inhibition of NF-κB activity by binding to IKK inhibitor binding site	[248]
		Prevention of NF-κB interaction with DNA	[249]
Quercetin	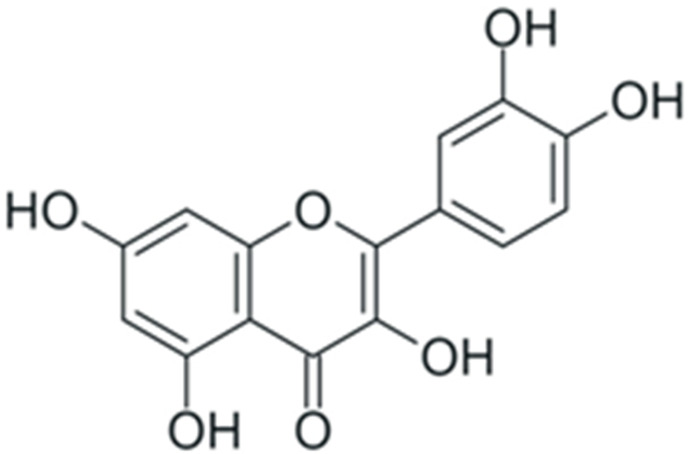	Downregulation of NF-κB p65 expression	[257]
		Upregulation of IκBα expression	[258]
		Downregulation of NF-κB expression, combined with upregulation of IκB expression	[259]
		Downregulation of NF-κB and IκB expression, reduction in IKK1/2 phosphorylation	[260]
		Downregulation of NF-κB p65 and p50 expression	[261]
		Prevention of upregulation of NF-κB expression	[262]
Resveratrol	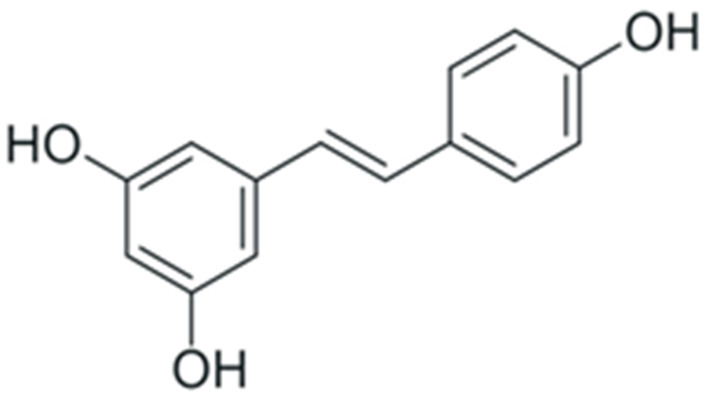	Decrease in NF-κB p65 expression	[263]
		Prevention ov NF-κB dimerization and nuclear translocation	[264]
		Downregulation of NF-κB expression	[265]
		Suppression of NF-κB activity	[266]
		Promotion of NF-κB-inhibiting activity of SIRT1	[267]
		Inhibition of NF-κB activation, combined with an increased secretion of IL-8	[268]
		Promotion of NF-κB expression and nuclear translocation, stimulation of IκB phosphorylation	[269]
Genistein	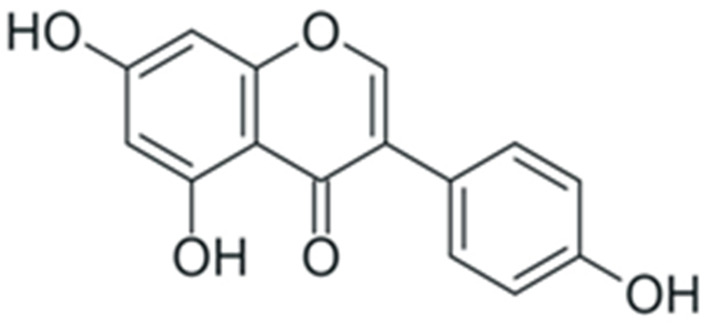	Suppression of NF-κB expression by upregulation of miR-29	[270]
		Downregulation of NF-κB expression	[271,272]
		Downregulation of NF-κB expression by suppression of Notch-1 activity	[273,274]
		Inhibition of IκB and NF-κB phosphorylation, prevention of NF-κB nuclear translocation	[275]
		Activation of IKK, increase in NF-κB phosphorylation (in combination with daidzein)	[276]
		Facilitation of NF-κB activation and nuclear translocation	[277]

**Table 4 ijms-24-10663-t004:** Antibacterial activity of plant polyphenols towards cancer-associated microorganisms.

Compound	Structure	Mechanism	Reference
Silibinin	Flavonol and phenolic acid heterodimer	Suppression of *H. pylori* growth	[293]
Kaempferol	Flavonol		[294]
(-)-epicatechin	Flavanol		
Ellagic acid	Tannin		[295]
Gallic acid	Phenolic acid		
Quercitin	Favonol		
Curcumin	Phenolic acid homodimer	Suppression of *C. jejuni* growth and prevention of *C. jejuni*-induced lesions	[300]
EGCG	Flavanol and phenolic acid heterodimer		[301]
Resveratrol	Stilbene		[302,303]
EGCG	Flavanol and phenolic acid heterodimer	Suppression of biofilm formation by *F. nucleatum*, combined with alleviation of inflammation induced by this microorganism	[304]
Theaflavines	Condensated flavanols		
Resveratrol	Stilbene		[305]

## Data Availability

Not applicable.

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
