# Peer review of "Inhibition of Cancer Development by Natural Plant Polyphenols: Molecular Mechanisms"

_ijms, 2023, doi:10.3390/ijms241310663_

Round 1
Reviewer 1 Report
Lubitelev and Studitsky summarised the molecular mechanisms involved in the inhibition of cancer development by natural plant polyphenols. The authors introduced how plan polyphenols act on key pathways in DNA damages, epigenetic regulation, tumour-promoting inflammation (especially the NF-kb pathway), and cancer-related microbiota.
Major comments:
Although the authors have already reviewed the aforementioned mechanisms of how tea polyphenols are involved in anti-cancer/cancer prevention, one key point is missing here. The authors mainly focus on the role of plant polyphenols on cancer cells (including how polyphenols impact DNA damage, epigenetic regulation, and the NF-kB pathway), however, they did not mention any literature on how plant polyphenols act on cancer-related immune cells. As immune cells are key components of the tumour microenvironment, the authors should review the progress of plant polyphenols on cancer-related immune cells.
In the section discussing the interaction of plant polyphenols with cancer-associated microbiota, the authors cited many studies that only report that polyphenols could inhibit the growth of bacteria in other diseases, not in cancers. Microbiota is usually linked with immune cell populations, and the authors should comment on how plant polyphenols interact with immune cells and then link them with microbiota.
The Introduction section should be reorganized. The authors provided excessive information on the process of malignant transformation, which is not used in the following sections. Instead, the authors should briefly start with an overview of the currently available therapies for cancer (including surgery, radiotherapy, immune therapy such as immune checkpoint inhibition, CAR-T cell therapy, etc.). Then, they can move on to discussing chemotherapy drugs and how plant polyphenols have been used in cancer therapy/prevention and their efficacy in cancer prevention and anti-cancer therapy.
In the manuscript, the authors mentioned various specific plant polyphenols such as quercetin, curcumin, EGCG, resveratrol, etc. Since the authors provided a classification of plant phenolic compounds in Figure 1, it would be better to include a summarized table at the end, showing the mechanisms in which these main categories of polyphenol compounds are involved in anti-cancer therapy. For example, the authors can summarize that phenolic acid compounds are involved in certain detailed mechanisms, while flavanol compounds are involved in other different mechanisms.
It is important to decipher, under each section, which structure is mainly responsible for certain mechanisms. This will help the authors better understand the links between compound structure and their actions."
Minor comments:
1. Line 365-369, some words stick to each other which makes it difficult to read. The authors should go through the manuscript carefully to revise these problems.
2. Missing of references, for example, line 109-111, line 277-279, line 349-352.
As suggested in my minor comments.
Reviewer 2 Report
In this research, the authors reviewed the recent development of inhibition of cancer development by natural plant polyphenols: molecular mechanisms. Generally, it’s meaningful and interesting review. In my opinion, the current version of this manuscript fits the scope of International Journal of Molecular Sciences and could be accepted after major revision.
My specific comments are in detail listed below:
1. All the figures are of low quality and black. In my opinion, high quality color figures may be better.
2. In this review, how natural plant polyphenols affect the anti-tumor immunotherapy should be added, especially its effects on PD-L1 expression. Some references should be added to this part including 10.1016/j.jconrel.2022.11.004.
3. Some minor mistakes existed in this paper. The authors should carefully check it.
4. How some other immune checkpoints was affected by natural plant polyphenols in solid tumors should be more clearly reviewed in this review if possible, including CTLA-4, VISTA, LAG-3, TIGIT and TIM-3.
5. In Line 936-943, it’s better to also point out some drawbacks of the chemical modifications of natural phenolic compounds.
6. Some tables may could be added to show the structures of typical natural plant polyphenols and its chemical derives.
7. How natural plant polyphenols affect the DNA damage repair process should be added to this review. Some references could be added including 10.1016/j.ijbiomac.2022.10.167.
8. The effects of natural plant polyphenols on T cell proliferation and T cell killing capacity could be also revealed in this review.
Reviewer 3 Report
Dear Authors. The presented work is a description of the mechanisms of anticancer activity of compounds from the polyphenol group. The presented work accurately and competently presents the current knowledge of the mechanisms of anticancer activity of these compounds.
Kindly provide some corrections:
1. In the introduction to the review, the concept of chemoprevention should be explained to the reader. Reading other publications I see that this term is often little known.
2. Figures 2 and 3 are prepared correctly, unfortunately their quality is not up to current standards. I recommend using the biorender tool or drawing it in color by a computer graphic designer.
3. each chapter should be accompanied by a table with a list of the compounds discussed, their mechanism of action and literature references. Such an amendment will improve the quality of the work and make it easier to read.
4. The last sentence is misleading. The main purpose of polyphenols will not be treatment, but prevention, possibly late chemoprevention. Many studies on compounds in this group including clinical trials confirm the effectiveness of dietary supplementation with polyphenols. This is especially seen in the Asian population.
I think that the conclusion of the paper should have a different meaning and definitely encourage a varied diet rich in polyphenols.
Round 2
Reviewer 2 Report
The current version of this manuscript could be accepted.
Author Response
We thank the reviewer for the positive comments
Reviewer 3 Report
Dear Authors.
Thank you for answering the questions and making the corrections. Accepts the review in its current version.
Author Response
We thank the revieweк for the positive comments